Analysing the spatial variation of soil respiration during the early growing season of different grasslands in China

Liu Jie 1 2
Huang Ni huangni@radi.ac.cn 1
Wang Li 1
Lin Xiaoyu 1 2
Zhu Luying 1 2
Niu Zheng 1 2
Zhang Yuelin 1
Duan Wensheng 1
1 Key Laboratory of Remote Sensing and Digital Earth, Aerospace Information Research Institute, Chinese Academy of Sciences , Beijing , China
2 College of Resources and Environment, University of Chinese Academy of Sciences , Beijing , China
Oehlmann Jörg
Electronic publication date: 2024 Nov 28
Publication date: 2024
Volume: 12
Electronic Location ID: e18480
Received 2024 Jan 25; Accepted 2024 Oct 16
Copyright: ©2024 Liu et al.
Copyright year: 2024
Copyright holder: Liu et al.
License: This is an open access article distributed under the terms of the Creative Commons Attribution License, which permits unrestricted use, distribution, reproduction and adaptation in any medium and for any purpose provided that it is properly attributed. For attribution, the original author (s), title, publication source (PeerJ) and either DOI or URL of the article must be cited.
License URL: https://creativecommons.org/licenses/by/4.0/

Keywords: Soil respiration, Early growing season, Grasslands, Spatial variation

Funding: National Natural Science Foundation of China 42371361 41771465 This work was supported by the National Natural Science Foundation of China (nos. 42371361 and 41771465). The funders had no role in study design, data collection and analysis, decision to publish, or preparation of the manuscript.

==============================
Background

As one of the most essential vegetation types, grasslands play a vital role in the global carbon cycle. However, current researches on the spatial variation (SV) of soil respiration (Rs) in grasslands faces great uncertainties.

Methods

The SV of Rs was analysed by obtaining Rs during the early growing season of three types of grasslands (i.e., alpine meadow, desert steppe, and typical steppe) and related impact factors at 19 sites.

Results

The results demonstrated that during the early growing season, the Rs of the alpine meadow was the highest, followed by the typical steppe and desert steppe. Moreover, soil temperature was the primary factor affecting the SV of Rs in desert steppe. In contrast, soil water content influenced the SV of Rs in typical steppe. This study increases our understanding of the SV of Rs during the early growing season of different grasslands. It provides an important reference for accurately estimating the SV of Rs in grasslands at various time scales.

Introduction

Soil respiration (Rs) is defined as the release of CO2 from the soil to the atmosphere. It also involves autotrophic respiration by plant roots and root-associated fungi and heterotrophic respiration by microorganisms in the soil (Jian et al., 2022; Lloyd & Taylor, 1994). Rs is the second-largest component of carbon flux in the carbon cycle of terrestrial ecosystems (Haaf, Six & Doetterl, 2021). Small changes in Rs might lead to larger changes in atmospheric CO2, significantly affecting global climate change (Bond-Lamberty et al., 2019; Jian et al., 2020; Haaf, Six & Doetterl, 2021). As one of the most essential vegetation types, grasslands cover one-fourth of the earth’s total land area (Barrow, 1995) and hold about 20% of the world’s soil carbon stock. Therefore, an in-depth investigation of grassland Rs is crucial for understanding the global carbon balance (Jägermeyr et al., 2014; Li et al., 2020b).

Various abiotic and biotic factors may drive grassland Rs (Baldocchi, Chu & Reichstein, 2018). Grassland Rs is regulated by abiotic factors, such as environmental factors (i.e., soil temperature (ST) (Cui et al., 2020) and soil water content (SWC) (Bani et al., 2018; Zhang et al., 2024)) and soil properties (i.e., soil carbon (Xiong, Wang & Sun, 2023) and nitrogen (Li et al., 2019)), which have been continuously discussed in recent decades. The biotic factors affecting grassland Rs primarily include vegetation type (Geoghegan, Langley & Chapman, 2021), root biomass (Sokol et al., 2018; Xiong, Wang & Sun, 2023) and litter (Badraghi et al., 2021). In addition, long-term vegetation cover, quantified by remotely sensed data, may also exert a significant regulatory impact on grassland Rs (Cavender-Bares et al., 2022). Therefore, grassland Rsmay exhibit spatial variation (SV) due to the spatial heterogeneity in biotic factors (i.e., canopy structure (Zheng et al., 2021), aboveground biomass (Xu et al., 2021), species richness (Kanga et al., 2023)), environment factors (i.e., air temperature and precipitation (Li et al., 2020a; Luo et al., 2022; Song et al., 2021b)), and soil properties (i.e., soil carbon (Xiong, Wang & Sun, 2023) and nitrogen (Li et al., 2019)).

Several studies have analyzed the SV of Rs in a single grassland type (Shi et al., 2020a; Zhao et al., 2017). However, few of them focused on the SV of Rs in multiple grasslands. For example, Geng et al. (2012) demonstrated that belowground root biomass is a key driver of SV of Rs during the peaking growing season of the alpine meadow. Fóti et al. (2008) discovered that SWC is the primary factor affecting the SV of Rs in semi-arid grasslands. Shi et al. (2019) found that grazing and N addition considerably affect the SV of Rs in meadow steppe. Different grasslands have various ecophysiological characteristics and the environmental responses to these characteristics may be different (Duan et al., 2021; Wang et al., 2015; Zhang et al., 2018). To gain insights into the spatial heterogeneity of grassland Rs, the SV of Rs in different grasslands must be analysed to increase our understanding of grassland Rs dynamics.

Previous studies also primarily focused on SV of Rs during the growing or peaking season (Liu et al., 2024; Qin et al., 2023). In contrast, there is little research on the SV of Rs during other periods, such as the early growing season. Several recent studies have revealed that the SV of grassland Rs exhibited temporal changes because of environmental factors (i.e., ST and SWC) (Fóti et al., 2018; Jian et al., 2021; Shi et al., 2020a). Except for the peak growing season, Rs during the early growing season accounted for a large portion of annual Rs (Ma et al., 2018).

Impact factors during the early growing season were apparently different from those at other periods, leading to the varying response of grassland Rs. For example, Wang et al. (2023) demonstrated ST exerted a greater impact on Rs during the early growing season than that on the late growing season in sandy grasslands. However, Yang et al. (2020) found that increased precipitation in the early growing season stimulates Rs more than in the late growing season in a semi-arid grassland. Until recently, there was no consensus on the major impact factors of grassland Rs during the early growing season. Therefore, it is critical to further investigate grassland Rs and its influencing factors during the early growing season. The temperate grassland in Inner Mongolia of northern China is a crucial region of the Eurasian steppe ecosystems and accounts for 12% of China’s total grassland area (Wang et al., 2017). The major grassland types are typical and desert steppes, accounting for 53.79% and 17.27% of total forage areas, respectively (Guo et al., 2021). In addition, the Tibetan Plateau is considered one of the most sensitive areas to climate change (An et al., 2021), half of which is covered by alpine grasslands (Yang et al., 2008). This study measured Rs and related impact factors during the early growing season of three types of grasslands (i.e., alpine meadow, desert steppe and typical steppe) in China. Next, the SV of Rs in these grasslands was analysed.

Material and Methods

Sampling sites

Site-level Rs during the early growing season of three types of grasslands (i.e., alpine meadow, desert steppe and typical steppe) in China were obtained in this study (Fig. 1A). The sampling sites of desert steppe and typical steppe were located in the Xilingol League, Inner Mongolia, which is in the central part of the Inner Mongolia Autonomous Region with latitudes 41°35′N–46°46′N and longitudes 111°09′E–119°58′E. This region has a temperate continental climate with cold winters and hot summers with an annual mean temperature of 1–4 °C and an annual mean precipitation of 150–400 mm. Grassland types primarily include desert steppe, typical steppe, and meadow steppe.

The alpine meadow site (37°36′N, 101°20′E) was located near the National Field Scientific Observatory for Alpine Grassland Ecosystems in Haibei, Qinghai (Fig. 1A). This site has an annual mean temperature of −1.9 °C and a mean yearly precipitation of 618 mm. In addition, this site has perennial snow and seasonal permafrost distribution. The major vegetation types are alpine meadow, alpine scrub and swampy meadow.

Measurement of Rs

A portable automatic soil carbon flux measurement system (Li-8100, Li-Cor Biosciences, Nebraska, USA) was used for measuring Rs. (Fig. 1B). In total, 18 sites were measured from May 3 to May 6, 2021, in the Xilingol League, Inner Mongolia. Eight were selected from the desert steppe, and ten were from the typical steppe. Three PVC rings with an inner diameter of 20 cm were placed for each plot. These rings were buried in the soil to a depth of three cm, keeping the aboveground height of each ring basically the same. Before each measurement, green plants in the rings were cut off to eliminate the impact of plant autotrophic respiration. The rings were placed 2 h before Rs measurement to minimize the effect of soil disturbance caused by the placement of the rings. Each measurement was made between 9:30 a.m. and 11:00 a.m. local time to ensure that plot-level measurements were spatially comparable.

Figure 1 Spatial distribution of sampling sites in three grassland types in China (http://211.159.153.75/) (A). The picture of the sample site in the alpine meadow in the Qinghai-Tibetan Plateau, China (B). Spatial distribution of sampling sites in desert and typical steppe across Inner Mongolia grasslands (C).

MNDVI is a multiyear (2000–2020) averaged normalised difference vegetation index (NDVI) from Landsat with a spatial resolution of 30 m (map data ©2024 Google).

At the alpine meadow site, a six-channel soil respiration measurement system (LI-8150, Li-Cor) was used to automatically and continuously measure Rs every hour (Fig. 1C). The Rs of six chambers were averaged to represent the Rsfor the site of alpine meadow site. Because there is only one site for the alpine meadow, the daily averaged Rs from the same period was selected as that of the Rs experiment in the Xilingol League (May 1–May 10, 2021) for comparative study.

Environmental factors and soil properties from field measurements

Rs was measured simultaneously at each site with environmental factors (i.e., ST and SWC) and soil properties (i.e., SOC, TN and total carbon (TC)). The ST was measured with the temperature probe connected with the LI-8100. SWC was measured using a soil moisture meter time-domain reflectometer. The soil samples were collected within the three rings when Rs measurements were completed for desert and typical steppes. However, the soil samples were collected outside the six rings for alpine meadow. Three 0–10 cm soil samples were collected. Then they were composited into one sample for soil property measurements (i.e., SOC, TN and TC). Specifically, SOC was measured using the potassium dichromate-endothermic method, and TN and TC were measured using the elemental analyser temperature combustion method.

Data from other sources

Because long-term biotic conditions may also affect the spatiotemporal variation of grassland Rs, multiyear (2000–2020) averaged normalised difference vegetation index (MNDVI) (Myneni et al., 1995) and land surface water index (MLSWI) (Jurgens, 1997) were calculated for all sampling sites. These calculations used remotely sensed surface reflectance data from Landsat with a spatial resolution of 30 m and a temporal resolution of 16 days based on an online geospatial data analysis cloud platform (Google Earth Engine, GEE) provided by Google.

Multiyear (2000–2020) averaged climate factors (i.e., mean air temperature and total precipitation) from ERA5 (https://cds.climate.copernicus.eu/cdsapp#!/dataset/reanalysis-era5-land-monthly-means) and the green vegetation index (i.e., normalised difference vegetation index (NDVI)) from Landsat were first calculated for each month in all sampling sites to characterize long-term monthly variations in hydrothermal and vegetation cover conditions of the three types of grasslands. Next, their monthly averages were calculated for all sampling sites in each type of grassland.

Remote sensed land surface temperature (LST) from Landsat and the soil moisture volume fraction from a Soil Moisture Active Passive L-Band radiometer on GEE were used for the representative analysis of sampling times and sites. Temperature-related factors were only considered to evaluate the representative sampling time. Then, the mean LST of all sampling sites was compared during the sampling time with that during the entire early growing season from 2000 to 2020. In contrast, temperature- and moisture-related factors were used to analyse the spatial representative of sampling sites by comparing LST and soil moisture in all sampling sites with that around the sample sites in each type of grassland.

Statistical analyses

The statistical analyses of Rs and related impact factors in three types of grasslands were performed using SPSS25.0 software (IBM, Armonk, NY, USA). One-way analysis of variance (ANOVA) was used to analyse the differences in Rs and impact factors among different grasslands. Because the alpine meadow contained only one sample site, the ANOVA analysis was conducted only for desert and typical steppes. To analyse the SV of Rs and related impact factors, their coefficients of variation (CV) were determined for the desert and typical steppes. Pearson’s correlation coefficients were also calculated to analyse the relationships between Rs and related impact factors. In addition, a regression model was constructed using stepwise regression to determine the crucial impact factors affecting the SV of Rs in desert and typical steppes. Origin2022 (OriginLab, Northampton, MA, USA) was used for graphing.

Results

Representative analysis of sampling times and sites

According to comparative analyses, the mean LST of all sampling sties in each type of grasslands during the sampling time was close to that during the entire early growing season from 2000 to 2020 (Fig. 2). The LST and soil moisture volume fraction of the sampling sites had near uniform distribution within these surrounding sites (Fig. 3) for three types of grasslands. The mean LST and soil moisture volume fraction of the sampling sites were close to their corresponding value from the surrounding sites for desert and typical steppes. These findings indicated that the sampling times and sites for each type of grasslands represented the entire early growing season and a large spatial region, respectively.

Figure 2 Distributions of the land surface temperature (LST) during the entire early growing season from 2000 to 2020 and the mean LST of all sampling sites during the sampling times in three types of grasslands.

Figure 3 (A-F) Distributions of the land surface temperature (LST) and soil moisture volume fraction of all sampling sites and their surrounding sites for the three types of grasslands.

Analysis of Rs and impact factors in different grasslands

Based on ANOVA results, significant (P < 0.05) differences were observed in Rs during the early growing season among the three types of grasslands (Fig. 4A). The alpine meadow had the highest Rs of 2.44 µmol m−2 s−1, followed by the typical steppe with mean Rs of 1.30 µmol m−2 s−1 and the desert steppe with mean Rs of 0.52 µmol m−2 s−1 (Fig. 4A). A minor difference in ST (P > 0.05 (i.e., not statistical significant), Fig. 4B) was observed among the three types of grasslands. In contrast, a significant difference was presented in SWC (P < 0.05, Fig. 4C). The SWC of the alpine meadow (40.32%) was higher than that of the desert and typical steppes. The mean values of SWC of the desert and typical steppes were 6.02% and 11.11%, respectively. Among the three types of grasslands, significant differences (P < 0.05) were observed in soil properties (i.e., SOC, TC, TN and soil carbon-to-nitrogen ration (C:N)) (Figs. 4D–4G) and MNDVI (Fig. 4H), but no difference was found for MLSWI (Fig. 4I).

Figure 4 Soil respiration and impact factors during the early growing season in three types of grasslands.

Rs is the soil respiration (A), ST is the soil temperature (B), SWC is the soil water content (C), SOC is the soil organic carbon content (D), TC is the soil total carbon content (E), TN is the soil total nitrogen content (F), C:N is the soil carbon-to-nitrogen ratio (G), MNDVI is multiyear averaged NDVI (H), and MLSWI is multiyear averaged LSWI (I).

SV of Rs during the early growing season of desert and typical steppes

Rs varied substantially during the early growing season across all sampling sites in desert or typical steppes based on the CV analysis of Rs (Fig. 5). Moreover, the CV of Rs during the early growing season of the typical steppe was larger than that of the desert steppe. Among all impact factors, the desert steppe demonstrated the largest CV of TC and the smallest CV of ST, whereas the typical steppe showed the largest CV of MLSWI and the smallest CV of C:N (Fig. 5). Desert steppe had a large CV in SOC and C:N, and the typical steppe exhibited large SV in other impact factors.

Figure 5 Coefficient of variation (CV) of soil respiration (Rs) and impact factors during the early growing season of two types of grasslands.

ST is the soil temperature. SWC is the soil water content. SOC is the soil organic carbon content. TN is the soil total nitrogen content. TC is the soil total carbon content. C:N is the soil carbon-to-nitrogen ratio. MNDVI is the multiyear averaged NDVI. MLSWI is the multiyear averaged LSWI.

Among all impact factors, Rs during the early growing season of desert steppe revealed the highest positive correlation with MNDVI and the highest negative correlation with SWC (Fig. 6A). Rs during the early growing season of the typical steppe only depicted a strong positive correlation with SWC (Fig. 6B). Stepwise regression results (Table 1) revealed that SWC was an essential predictor of SV of Rs during the early growing season of the typical steppe (R2 = 0.88, P < 0.05). However, ST was important in explaining the variables for SV of Rs during the early growing season of the desert steppe.

Figure 6 The relationships between soil respiration (Rs) and various impact factors during the early growing season of two types of grasslands.

Pearson’s correlation coefficients between Rs and impact factors during the early growin. ST is the soil temperature. SWC is the soil water content. SOC is the soil’s organic carbon content. TN is the soil’s total nitrogen content. TC is the soil’s total carbon content. C:N is the soil’s carbon-to-nitrogen ratio. MNDVI is a multiyear averaged NDVI. MLSWI is a multiyear averaged LSWI.

Discussion

Differences between Rs and related impact factors during the early growing season in different grasslands

Rs during the early growing season varied significantly among the three types of grasslands. This result might be attributed to the complicated interactions of various impact factors, such as vegetation type, soil properties and site-specific climatic conditions. Desert and typical steppes had lower Rs than alpine meadows, which were consistent with previous studies (Feng et al., 2018; Wang et al., 2020). The long-term lower air temperature, higher precipitation and growing-season NDVI at the alpine meadow (Fig. 7) caused more carbon fixed by photosynthesis to be retained in the soil and thus led to the accumulation of SOC (Deng et al., 2023; Jia et al., 2019). The abundant SOC provided a sufficient source substrate for Rs. Furthermore, alpine meadows exhibited the highest vegetation cover during the early growing season, which may contribute to the highest Rs during the early growing season of the three types of grasslands (Soong et al., 2020).

As the transition zone between the typical steppe and the desert, desert steppe generally has higher air temperature, lower precipitation and vegetation cover (i.e., NDVI) (Fig. 7). It was the driest among the three types of grasslands based on the long-term seasonal variation of monthly total precipitation (Fig. 7). Its dominant vegetation are shrubs and herbs, with sparse vegetation cover and low aboveground biomass (Angerer et al., 2008; Gao et al., 2022). Furthermore, long-term human grazing activities have led to severe soil sanding and low SOC in the desert steppe (Song et al., 2017; Song et al., 2021a). During the early growing season, the low SOC may restrict the diversity and activity of soil microorganisms and thus generate low Rs in the desert steppe (Lee et al., 2021; Zhou et al., 2013). The annual precipitation and growing-season NDVI of the typical steppe were higher than those of the desert steppe (Fig. 7). This activity may cause aboveground biomass and vegetation cover of typical steppe two times higher than those of the desert steppe during the same period (Jägermeyr et al., 2014; Tang et al., 2013), which may contribute to a richer SOC. Moreover, a higher Rs was exhibited in typical steppe than in desert steppe (Fig. 4).

Factors affecting the SV of Rs during the early growing season of the desert and typical steppes

During the early growing season of the desert steppe, ST was found to be the primary factor affecting the SV of Rs based on step regression analysis (Table 1). This result was inconsistent with recent studies demonstrating that soil moisture considerably affected the Rs of desert steppe (Gao et al., 2022; Suseela et al., 2012; Zhang et al., 2021). This inconsistency might be attributed to the correlations among various variables in the desert steppe (Fig. 6A). MNDVI and MLSWI had a strong positive correlation with Rs, and ST showed a strong positive correlation with both vegetation indices. Previous studies had clearly confirmed a significant positive correlation between NDVI and aboveground biomass of grasslands (Spehn et al., 2000; Eisfelder, Kuenzer & Dech, 2012). In addition, SOC of grasslands primarily comes from the carbon fixed by vegetation photosynthesis. Therefore, higher MNDVI indicates that the desert steppe had relatively higher SOC promoting Rs. The negative correlation between MNDVI and SWC could partly explain the negative correlation between SWC and Rs during the early growing season of the desert steppe.

In contrast, SWC was an essential factor influencing the SV of Rs in the typical steppe. The typical steppe in this study was distributed in the arid and semi-arid regions of China, where soil moisture is a crucial limiting factor of various ecosystem processes (Jia, Zhou & Yuan, 2007; Li et al., 2021). Shi et al. (2020b) discovered that soil water characteristics limited the soil CO2 release rate during the dry season. Increasing soil water availability, which promotes microbial community and fine root activity (Berry & Kulmatiski, 2017), accelerates substrate diffusion and soil organic matter decomposition (Suseela et al., 2012), thus increasing soil CO2 emission. Therefore, the significant effect of SWC on the SV in Rs during the early growing season of the typical steppe is entirely expected. However, the same phenomenon did not occur in the desert steppe, probably because the factors affecting Rs changes during the early growing season were not independent. For example, the MNDVI and ST correlated with the SWC (Fig. 6A), which partly weakened the effect of SWC on Rs in the desert steppe.

Based on field observations of Rs in 42 sample plots in the alpine meadow on the Tibetan Plateau, a previous study discovered that the SV in Rs during the peaking growing season of alpine meadow was affected by the belowground root biomass rather than the SWC (Geng et al., 2012; Huang, He & Niu, 2013). The inconsistency between this finding and our study might be due to the difference in the study time (mid-growing season vs. early growing season). This temporal difference would have led to changes in the proportion of the two major components of Rs in grassland (i.e., soil autotrophic respiration and heterotrophic respiration) and their responses to impact factors (Li et al., 2010; Nissan et al., 2023; Shi et al., 2022). The surface vegetation is sparse during the early growing season compared to the mid-growing season. Therefore, Rs was dominated by soil heterotrophic respiration during the early growing season than soil autotrophic respiration (Li et al., 2018; Yang et al., 2020).

Table 1 Step regression analysis for the relationships between soil respiration and impact factors during the early growing season of the desert and typical steppes.

Grassland type	R 2	Explanatory factors (weights)	
Desert steppe	0.91	ST (0.95)	
Typical steppe	0.88	SWC (0.94)	

Figure 7 Monthly averaged air temperature, total precipitation, and normalised difference vegetation index (NDVI) of the three types of grasslands from 2000 to 2020.

DS is the desert steppe, TS is the typical steppe, and MM is the alpine meadow. The red line indicates the time of this study.

Conclusions

This study analysed the differences in Rs during the early growing season among the three types of grasslands (i.e., alpine meadow, desert steppe, and typical steppe) based on site-level measured Rs and related impact factors. Alpine meadow had the highest Rs during the early growing season. The significant differences in Rs among the three types of grasslands may be attributed to impact factors. ST was the most crucial factors affecting the SV of Rs during the early growing season of the desert steppe. In contrast, SWC had the greatest effect on the SV in Rs during the early growing season of the typical steppe. In future studies, it will be necessary to construct prediction models by time periods to estimate Rs in grasslands accurately at different time scales.

Supplemental Information

Data S1 The raw data of the sample point

(A) grassland type (B) soil respiration (C) lalitude (D) longitude (E) soil temperature (F) soil water content (G) soil organic carbon (H) total nitrogen (I) total carbon (J) C:N

Additional Information and Declarations

Competing Interests

Author Contributions

Data Availability

The authors declare there are no competing interests.

Jie Liu performed the experiments, analyzed the data, prepared figures and/or tables, authored or reviewed drafts of the article, and approved the final draft.

Ni Huang conceived and designed the experiments, performed the experiments, authored or reviewed drafts of the article, and approved the final draft.

Li Wang conceived and designed the experiments, authored or reviewed drafts of the article, and approved the final draft.

Xiaoyu Lin analyzed the data, authored or reviewed drafts of the article, and approved the final draft.

Luying Zhu performed the experiments, prepared figures and/or tables, and approved the final draft.

Zheng Niu conceived and designed the experiments, authored or reviewed drafts of the article, and approved the final draft.

Yuelin Zhang performed the experiments, prepared figures and/or tables, and approved the final draft.

Wensheng Duan performed the experiments, prepared figures and/or tables, and approved the final draft.

The following information was supplied regarding data availability:

The raw measurements are available in the Supplementary File.

Map of China from http://211.159.153.75/.

The total precipitation and air temperature are available at ERA5: https://cds.climate.copernicus.eu/cdsapp#!/dataset/reanalysis-era5-land-monthly-means?tab=overview.

The soil moisture volume fraction is available from SMAP: https://developers.google.com/earth-engine/datasets/catalog/NASA_SMAP_SPL3SMP_E_005#bands.

The Landsat7 data is available at: https://developers.google.com/earth-engine/datasets/catalog/LANDSAT_LE07_C02_T1_L2.

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
