# Peer review of "Analysing the spatial variation of soil respiration during the early growing season of different grasslands in China"

_PeerJ, doi:10.7717/peerj.18480_

## Round 0.1 · original submission · Major Revisions

The reviewers recommended rejection and significant revisions for this manuscript. However, I believe it merits an opportunity for improvement. Both the introduction and discussion sections require a more coherent structure. The reviewer highlighted the presence of extraneous information in the introduction that detracts from the focus on the study's objectives. Additionally, the discussion should extend beyond reiterating the introduction to effectively interpret the findings. It is also important to clarify the rationale for selecting MNDVI and LSWI in this study. I concur with the reviewer’s observations and suggest that you meticulously address these comments. A diligently revised manuscript could be suitable for consideration in PeerJ.

Reviewer 1 ·

Basic reporting

discussions sections need more details and explain the evidence more clearly instead of rewriting most of the findings which are written in the results sections.

Experimental design

no comment

Validity of the findings

The methodology used by the authors is convincing, however, questions arise about the representative sampling because the samples were taken within a short period from May 3 to 6. Only one time throughout the early growing period.

Additional comments

The authors described that there are limited research on soil respiration during early growing seasons in different types of grassland ecosystems and provided a good introduction. The methodology used by the authors are convincing methods however questions arise about the representative sampling because the samples were taken within a short period from May 3 to 6 only one time throughout the early growing period. Authors should justify whether that sampling period is representative of early growing season and whether only one sampling at each location can explain spatial variability. Furthermore, your discussion sections need more details and need to explain the evidence more clearly instead of rewriting most of the findings which are written in results sections. See the comments for each section below:

Introduction
Line 61: accounted for most of the annual of Rs, of may not be necessary.
Line 41 and 42, 51 and 52, 68 and 69 are repeated which are mainly focused on why this research is important. I suggest you can include that sentences after you explained all of the limitations from previous research.
Materials and Methods
Line 87: -1.9 °C
Line 101 and 102: For how many times Rs was measured hourly? You only mentioned every hour.
Line 111 and 112: I will say composite if that represents a composite sample.
Line 119: Should be “To characterize the long-term vegetation cover and water conditions of these sites, “
Results
Line 143: significant differences in Rs…
Include P value.
Line 146 and 147 : Your P > 0.05 and you have states minor difference in ST. Please explain what does this term “minor differences” indicate.
Line 165 and 166: Please justify how SWC, SOC, TC and TN are all similar to Rs. The CV for SC, SOC and TN have distinct differences. If you are talking about trends where desert has lower CV than typical steppes. Please elaborate clearly.
Line 170 and 171: Comparison is between the desert and typical steppes so it should be two grasslands instead of three.
Line 174 and 179: Are you discussing figure 2 or figure 3?
Discussion:
Line 191 and 192: The statement is the repetition of sentence in line 143 and 144. Please rewrite it or focus on discussion rather than repetition of results.
Line 197 to 200: There is repetition of results sections again. I would recommend this paragraph needs further revision and delve more into discussions instead of rephrasing results.
Line 204 to 219: Here I see repetition of results sections again and rewrite these sections comparing the results from previous literatures. It should be okay to discuss results from literature with different scenarios as well.
Line 271: may be.

Reviewer 2 ·

Basic reporting

The authors of “Analyzing spatial variation of soil respiration during the early growing season of different grasslands in China “ cover an interesting topic with potentially global relevance, because the spatial variability of soil respiration may be a key factor in carbon balance estimation uncertainty. Moreover, grasslands cover large areas on Earth. This kind of vegetation, at least in semi-arid, arid lands, under natural, semi-natural conditions, is known for its large below-ground biomass and biological activity. However, the manuscript is not acceptable in its current form for publication in PeerJ not only because it is based on a minimal dataset, but also because there are problems with the structure of the introduction and the discussion, the context for the study is not presented appropriately and the figures and the single table show very little about the targeted field.

Experimental design

The Introduction collects a lot of necessary information related to the topic of the manuscript, but I miss clearer structure and arguments. You don’t give evidence about the “great potential” (L40-41) of grasslands to sequester carbon. Why is this true? Because of their large root system? Because of their species composition and phenology? There can be many reasons depending on the system.
I think that your grouping of single grassland type – multiple grasslands don’t assess an essential way of studying soil respiration spatial variability. I would consider collecting the introductory studies according to e.g., the complexity of the system in question. Like, there is already a heterogeneity without any vegetation (cf. works of e.g., Michael Herbst and Alexander Graf). There are causes for this (surface, SOC, ST, SWC…). A single-species crop vegetation reflects these, but also, and I lack this from your introduction, use fresh assimilates from GPP with all the consequences related to the physiological functional heterogeneity, canopy structure, etc. Grasslands with low or high species numbers are increasingly more complex, and those are your systems, but you don’t mention, at least briefly, how complex they are, etc., just, that I don’t understand here: “various ecophysiological characteristics”. Which one? Why?
A third problem I would address here is the question of choosing the early growing season in your study. If you admit that “peak growing season” accounts for most, why not give at least some ratio of the early season’s Rs activity from a cumulative one? I’m not entirely sure that your selected period is worth your effort, so please give more evidence.
Finally, there are no research questions, as I miss a better-structured introduction, I also miss the relevant knowledge gaps, there are no hypotheses to answer, so please, think over your research purposes again and provide these logically and concisely.
The Methods seem to be relevant to the study, but I think that a few days of measurement in one single year would give a very poor representation. Even the early growing seasons may differ from year to year, they depend on e.g., the previous year’s environmental conditions as well as the actual precipitation, temperature, etc., so there is a huge yearly difference that is not captured at all. Therefore, we cannot be sure that your experiment captured the variation you would like to capture and describe, of course, the spatial variation is more expressed or less, depending on the actual conditions, as you may know.
Furthermore, there is a new element here, the data acquisition of remotely sensed variables. I was missing the biological influence of soil respiration from the introduction, so please, complete the introduction according to my suggestions, and this part of the methods will not be a surprise anymore. And please be clearer about the usage of the 16 days and the monthly data. Why and for what did you use the one or the other? Please, describe the relevance of MNDVI and LSWI. What are they used for?
The same holds for the analysis of SOC, TN, TC. Why? Why didn’t you talk about the importance of these properties while introducing your topic? Please, reconsider the introduction related to this as well.
The data analysis fits your aims, but I have concerns about the comparability of the data from the sites. The data coverage is very different (7 sites, 11 sites, 1 site; 3 plots, 6 plots…) How would you compare those?
A simple CV is a very simple way of comparing variables “spatially”. It is highly dependent on the actual average and standard deviation, mean values close to zero give a very high CV without really differing from other variables. I suggest rescaling (standardizing) your variables first to overcome this problem.

Validity of the findings

The results are very simple, but the experiment didn’t allow you to assess more from the data, I admit. I’m not sure that, as I was suggested before, such a different data coverage per site would allow a meaningful comparison (cf. one line for the alpine meadow in Fig. 1., as an average from 6 replicates, 7*3 and 11*3 replicates averaged for the others?).
Otherwise, I’m not entirely convinced that your data are enough for any deep analysis in their actual form.

Additional comments

Minor points:

Abstract
You should rewrite the whole manuscript, if it’s possible so the abstract would probably be different, I have no suggestions now.

Introduction
L35: not just roots, but root-associated fungi
L49: …ecophysiological characteristics, the environmental…
L58: instead of “time” changes please use temporal changes or temporal variability.
L59: soil properties are not temporally variable characteristics, at least not at the same timescale as the others
L71: why those types of grasslands? Are they the most common? Do they cover a range of grassland types?
L73: a detailed analysis of

Methods
L120: (MNDVI)
L121: (MLSWI)

Results
L143-157: are these results based on the ANOVA? Please, mention this.
L158-161: the figure legend does not contain sub-plot references (a, b, …). Please, correct.
L174: I would suggest grouping your variables in a more meaningful way, MNDVI is not an environmental factor, please, be coherent with this throughout your manuscript. You may have soil variables, environmental factors, and biotic factors, e.g.
L177: is this related to Tab. 1?

Figures and tables
Tab. 1: for the desert steppe you mention MLSWI but MNDVI is presented as the most important factor.

Discussion - Conclusion
You cannot put a figure into the discussion. Please, start you results with presenting your actual Fig. 4 first with the comparison of the precipitation, temperature and NDVI of the three sites.
Otherwise, as I had major problems with the methods and the poor representation of you data, I will not read the discussion and conclusion, as I’m not entirely convinced that your data are enough to any deep analysis in their actual form.

---

## Round 0.2 · accepted · Accept

Thank you for the thorough revision of the manuscript and your patience in the review process. I have been asked to take over as the prior Academic Editor was unavailable. I hereby certify that you have adequately taken into account the reviewers' comments and improved the manuscript accordingly. Based on my assessment as an Academic Editor, your manuscript is now ready for publication. There are still a few grammatical errors that need to be corrected, but this can be done as part of the proofreading process.

Reviewer 1 ·

Basic reporting

The manuscript has significant improvements after the revisions. The organization of the manuscript is improved, and the previous comments were addresses satisfactorily.

Experimental design

It is good to see the additional analysis and incorporated data from other sources. The justification provided are acceptable and well explained.

Validity of the findings

The authors had justified their findings with adequate discussions.

Additional comments

There are many grammatical errors at several places. Sentences should not start with Abbreviations, it needs to be corrected. There are other errors such as space, commas and parenthesis.
Figure 3 seems blur and not clear and need refinement.

Reviewer 2 ·

Basic reporting

No comment.

Experimental design

No comment.

Validity of the findings

No comment.

Additional comments

No comment.